# Contribution of Autophagy to Epithelial Mesenchymal Transition Induction during Cancer Progression

**DOI:** 10.3390/cancers16040807

**Published:** 2024-02-16

**Authors:** Raffaele Strippoli, Reyhaneh Niayesh-Mehr, Maryam Adelipour, Arezoo Khosravi, Marco Cordani, Ali Zarrabi, Abdolamir Allameh

**Affiliations:** 1Department of Molecular Medicine, Sapienza University of Rome, 00161 Rome, Italy; raffaele.strippoli@uniroma1.it; 2National Institute for Infectious Diseases “Lazzaro Spallanzani”, I.R.C.C.S., 00149 Rome, Italy; 3Department of Clinical Biochemistry, Faculty of Medical Science, Tarbiat Modares University, Tehran P.O. Box 14115-331, Iran; r.niayeshmehr@modares.ac.ir; 4Department of Clinical Biochemistry, School of Medicine, Ahvaz Jundishapur University of Medical Sciences, Ahvaz 61357-15794, Iran; adelimaryam@rocketmail.com; 5Department of Genetics and Bioengineering, Faculty of Engineering and Natural Sciences, Istanbul Okan University, Istanbul 34959, Türkiye; arezoo.khosravi@okan.edu.tr; 6Department of Biochemistry and Molecular Biology, Faculty of Biological Sciences, Complutense University of Madrid, 28040 Madrid, Spain; mcordani@ucm.es; 7Instituto de Investigaciones Sanitarias San Carlos (IdISSC), 28040 Madrid, Spain; 8Department of Biomedical Engineering, Faculty of Engineering and Natural Sciences, Istinye University, Istanbul 34396, Türkiye; ali.zarrabi@istinye.edu.tr; 9Department of Research Analytics, Saveetha Dental College and Hospitals, Saveetha Institute of Medical and Technical Sciences, Saveetha University, Chennai 600077, India

**Keywords:** autophagy, epithelial mesenchymal transition, cell death, cell adhesion molecules, cell proliferation, differentiation

## Abstract

**Simple Summary:**

This manuscript focuses on the complex relationships between autophagy and epithelial mesenchymal transition (EMT) in cancer. Autophagy, a cellular degradation process, and EMT, a mechanism where epithelial cells acquire mesenchymal features, both play significant roles in cancer development. This review aims to explore how these processes interact, particularly how autophagy impacts cancer cell fate during EMT. The findings from this study are expected to contribute to a better understanding of cancer biology and could potentially impact cancer treatment strategies, as both autophagy and EMT are considered targets for therapy.

**Abstract:**

Epithelial Mesenchymal Transition (EMT) is a dedifferentiation process implicated in many physio-pathological conditions including tumor transformation. EMT is regulated by several extracellular mediators and under certain conditions it can be reversible. Autophagy is a conserved catabolic process in which intracellular components such as protein/DNA aggregates and abnormal organelles are degraded in specific lysosomes. In cancer, autophagy plays a controversial role, acting in different conditions as both a tumor suppressor and a tumor-promoting mechanism. Experimental evidence shows that deep interrelations exist between EMT and autophagy-related pathways. Although this interplay has already been analyzed in previous studies, understanding mechanisms and the translational implications of autophagy/EMT need further study. The role of autophagy in EMT is not limited to morphological changes, but activation of autophagy could be important to DNA repair/damage system, cell adhesion molecules, and cell proliferation and differentiation processes. Based on this, both autophagy and EMT and related pathways are now considered as targets for cancer therapy. In this review article, the contribution of autophagy to EMT and progression of cancer is discussed. This article also describes the multiple connections between EMT and autophagy and their implication in cancer treatment.

## 1. Introduction: The Fate of Normal and Transformed Cells

The concept of cell fate pertains to the prospective identity of a cell or its progeny [1]. Distinct cell fates play a critical role in tissue development, organismal homeostasis maintenance, and response to environmental disturbances [2]. Basically, a normal cell can manifest one of five distinct cell fate types: proliferation, differentiation, quiescence (G0 phase), senescence, and cell death (regardless of the mechanism of cell death). Complex regulatory mechanisms, resulting from the interplay between intracellular regulatory networks and external environmental stimuli, strongly govern cell fate decisions. These decisions often occur in the G1 phase of the cell cycle [3,4,5].

Cellular transformation is caused by initial genetic alterations, which are followed by subsequent changes that progressively drive the cell towards a malignant phenotype. Cancer cells emerge as transformed derivatives of normal cells, exhibiting an extraordinary capacity for rapid proliferation and dedifferentiation. This transformation entails the progressive accumulation of DNA mutations in cancer-related genes, predominantly occurring within proto-oncogenes and/or tumor suppressor genes [6,7]. These mutations can ultimately lead to alterations in the cell’s morphology, as well as its biochemical and molecular characteristics. In such cases, cells acquire the capacity to escape apoptosis, develop resistance to anti-growth signals, achieve unlimited replicative potential, and maintain continuous angiogenesis. Under these circumstances the cells can acquire the capacity to invade neighboring tissues and metastasize [8]. Consequently, cancer-associated modifications have the potential to perturb cellular fate decisions by impacting essential cellular processes such as proliferation, differentiation, metabolism, and cell death [9,10]. Exogenous and endogenous agents, including ionizing radiation, chemical carcinogens, oncoviruses, tobacco and alcohol consumption, as well as disorders that predispose individuals to DNA damage and cancer, are contributing factors [6,11].

When cells encounter DNA damage, they activate a network of intracellular pathways to repair the damage and restore DNA integrity, collectively known as the DNA damage response (DDR) [12]. Failure to repair DNA lesions can result in the activation of cell death processes, primarily apoptosis, as the main pathway of cell death. If these pathways are inactivated, they can contribute to genomic instability [13]. Ataxia telangiectasia mutated (ATM) and ataxia telangiectasia and RAD3-related (ATR), belonging to the PIKK (phosphoinositide 3-kinase related kinase) family, are recognized as key regulators of the DDR [14]. Important functions of ATM and ATR include recognition of DNA damage and regulation of DNA repair, cell cycle checkpoints, and apoptosis signaling pathways [15]. ATM predominantly responds to DNA double-strand breaks (DSBs), while ATR functions as a sensor that responds to a wider range of DNA damages, including DNA single-strand breaks (SSBs), DSBs, and DNA lesions during replication [14,16]. Some similar phosphorylation targets have been identified for both pathways. Upon activation of ATM/ATR kinases, a signaling cascade is initiated, transmitting signals to downstream mediators such as checkpoint kinase 1/2 (CHK1/CHK2), cell division cycle 25 (CDC25) protein, and cyclin-dependent kinase (CDK). Subsequently, DDR pathway proteins like breast cancer type 1/2 susceptibility protein (BRCA1/2) and p53 can undergo phosphorylation [14]. Evidence indicates that autophagy can play a significant role in the DDR pathway [17]. Studies have demonstrated that DNA damage can trigger autophagy, with various mechanisms attributed to this phenomenon [18]. For instance, certain DNA damage inducers have been found to induce autophagy as well. One example is the activation of the ATM pathway, which links DDR to autophagy induction by activating the AMP-activated protein kinase (AMPK) pathway and the transcription factor forkhead transcription factor O subfamily member 3a (FOXO3a) that is involved in the regulation of ATM for DNA repair. AMPK is capable of counteracting the inhibitory effect of mammalian target of rapamycin complex 1 (mTORC1) on the autophagy pathway through phosphorylation of tuberous sclerosis complex (TSC2). Moreover, AMPK can promote autophagosome formation by activating unc-51-like autophagy activating kinase 1 (ULK1). Additionally, studies have shown that FOXO3a plays a role in the transcription of genes related to autophagy [13,14].

Apoptosis and autophagy are important factors to control the balance of cell proliferation and death and to save the organ from uncontrolled cell proliferation. Therefore, the autophagic process causing intercellular lysosomal degradation is an important phenomenon which contributes to cell fate decisions [19,20]. The impact of autophagy on the fate of normal and cancer cells is mediated by different mechanisms. Lysosomal degradation, cell survival, apoptosis induction, senescence and aging, regulation of cell differentiation, and modulation of the carcinogenesis process can happen in response to autophagy [21,22,23,24]. Specific outcomes of autophagy in cancer cells depend on cell type or context, cellular conditions, stage of carcinogenesis, and interactions with other cellular processes and signaling pathways [25]. Since the EMT process is recognized as a dedifferentiation process involved in cancer progression, this study aims to investigate the role of autophagy and EMT processes in the development and treatment of cancer, as well as explore different aspects of the interaction between EMT and autophagy.

## 2. Autophagy and Cancer

### 2.1. Role of Autophagy in Cancer Development and Progression

Macroautophagy, also known as autophagy, recycles damaged or unnecessary components within cells. It involves initiation, phagophore formation, recognition of cargo, fusion between autophagosome and lysosome, degradation of cargo, and autophagolysosome breakdown. When cells experience stress or nutrient deprivation, a phagophore forms near the target cargo and expands to become an autophagosome. Cargo receptors like sequestosome 1 (SQSTM1)/p62 identify and recruit the cargo, which is then degraded by lysosomal enzymes in the autophagolysosome. This breakdown releases recycled molecules back into the cell’s cytoplasm [26,27].

Autophagy is regulated by nutrient and energy-responsive signaling pathways. For instance, mTOR (mammalian target of rapamycin) and Class III phosphoinositide 3-kinase (PI3K)/Akt pathways inhibit autophagy through protein phosphorylation, while the AMP-activated protein kinase (AMPK) pathway promotes autophagy [26]. ULK1/2, crucial for autophagy induction, can be regulated by mTOR and AMPK. AMPK can induce ULK activation by phosphorylating specific sites, such as S317 and S777. Additionally, mTOR can inhibit autophagy by phosphorylating ULK, including the site S757, thereby inhibiting the interaction between ULK and AMPK [28] (Figure 1). Transcription factors (TFs) like TFEB control the expression of lysosomal and autophagy-related genes such as ATG9B, LC3, and SQSTM1, and forkhead box protein O1 (FOXO1) activates autophagy-related genes such as LC3b, Gabarapl1, PI3kIII, Ulk2, Atg12 l, Beclin1, Atg4b, and Bnip3. Additionally, post-translational modifications like phosphorylation, acetylation, and ubiquitination regulate protein activity and stability in autophagy [26,29].

The role of autophagy in cancer is complex, serving both as a tumor suppressor and a pro-survival mechanism (Figure 2). Autophagy’s dual role is evident in cancer progression. It acts as a tumor suppressor at early stages, preventing the accumulation of damaged proteins and organelles, promoting cell death, and hindering cancer cell survival. However, at later stages, especially with oncogenic K-RAS pathway activation, autophagy shifts to a pro-survival mechanism, aiding cancer cells in resisting stress conditions like nutrient deprivation, hypoxia, and chemotherapy-induced apoptosis. This pro-survival function can contribute to tumor growth and therapy resistance. Additionally, autophagy can play a role in degrading the extracellular matrix, promoting cancer cell motility, and facilitating tumor cell invasion and metastasis [22,30].

Some studies have reported that defects in autophagy can contribute to tumorigenesis by promoting genomic instability, inflammation, and metabolic dysregulation [31,32,33].

Autophagy performs several functions that help stabilize the genome, including maintaining mitochondrial quality, reducing the accumulation of reactive oxygen species (ROS), defending against carcinogenic pathogens, breaking down misfolded or overexpressed proteins, and removing micronuclei and damaged nuclear parts. Autophagy also influences the DNA damage response (DDR) and the processing of genomic lesions. Evidence suggests that autophagy is closely linked to the DDR network, and that DNA damage can stimulate autophagy in both pathological and nonpathological situations. The proper functioning of the DNA repair machinery is contingent upon a highly efficient autophagic process. Autophagy, in this context, plays a pivotal role in DNA repair. One crucial aspect is its contribution to maintaining bioenergetic fitness. Autophagy provides metabolic precursors essential for generating ATP. This ATP generation is particularly critical for supporting optimal DNA repair processes. Moreover, autophagy actively participates in ensuring nucleotide homeostasis, a fundamental requirement for DNA synthesis. By regulating the availability and balance of nucleotides, autophagy contributes to the integrity of the cellular genome, facilitating the accurate and efficient synthesis of DNA molecules [23,24]. According to Delaney et al., monoallelic loss of Beclin-1 (BECN1) has been shown to promote genomic instability in ovarian cancer. Knocking down BECN1 and microtubule-associated protein 1 light chain 3 beta (MAP1LC3B/LC3B), two frequently deleted autophagy genes in ovarian cancer, led to genomic instability and increased migration rates in atypical ovarian cancer cells. In a mouse model, haploinsufficiency of BECN1 was observed to allow and enhance genomic instability in ovarian cancer. These findings suggest that autophagy deficiency, particularly monoallelic loss of BECN1, can play a role in genome instability in cancer [25]. Nevertheless, in cases of excessive or prolonged DNA damage, autophagy eventually induces senescence or cell death, thereby inhibiting the proliferation of cells harboring genomic aberrations [33,34].

Inflammation as a hallmark of cancer has been shown to be involved in various stages of tumor progression. Autophagy has been implicated in regulating inflammation in cancer by promoting the degradation of damaged cellular components and limiting the release of pro-inflammatory signals. However, the relationship between autophagy and inflammation in cancer is complex and context-dependent [35]. In addition, autophagy can enhance the processing and presentation of tumor antigens, which stimulates anti-tumor immunity. However, cancer cells may reduce autophagy to evade immune surveillance [36]. Mommersteeg et al. investigated the role of autophagy in inflammation and the development of Helicobacter pylori-associated gastric cancer. They found that this bacterium activates autophagy, which can be dysregulated by a single nucleotide polymorphism in the autophagy gene ATG16L1, leading to increased endoplasmic reticulum stress and inflammation. The study suggests that dysregulation of autophagy and inflammation pathways by this genetic variation could contribute to the development of this type of cancer [37].

Moreover, the crosstalk between autophagy and metabolism in cancer is bidirectional and complex. Autophagy plays a crucial role in regulating cellular metabolism, and metabolic changes in cancer cells can also influence autophagy. Jiao et al. indicated that autophagy plays a role in regulating glycolytic metabolism in liver cancer cells by targeting and breaking down hexokinase 2 (HK2). The degradation process involves the ubiquitination of HK2 facilitated by the E3 ligase TNF receptor-associated factor 6 (TRAF6) and subsequent recognition by the autophagy receptor protein SQSTM1/p62. Both in vitro and in vivo experiments demonstrated that 3-bromopyruvate, a pyruvate analog targeting HK2, decreased the growth of tumors with impaired autophagy. This study suggests that targeting glycolysis through the TRAF6- and SQSTM1-mediated ubiquitination system may be a potential therapeutic intervention for autophagy-impaired liver cancer [38]. Moreover, Perera et al. revealed that the induction of autophagy in pancreatic ductal adenocarcinoma (PDA) is a component of a broader transcriptional program. This program, mediated by the MiT/TFE family of transcription factors (TFs), orchestrates lysosome biogenesis and function, as well as nutrient scavenging, playing a crucial role in upregulating the expression of a network of genes, promoting elevated levels of lysosomal catabolic function crucial for the growth of PDA. Metabolite profiling indicates that MiT/TFE-dependent activation of autophagy–lysosome pathways is specifically essential for sustaining intracellular amino acid pools. These findings highlight the crucial role of autophagy in regulating the metabolism in pancreatic cancer and identify transcriptional activation of clearance pathways as a newly identified characteristic of highly aggressive malignancy [39].

### 2.2. Role of Autophagy in Cancer Stem Cell Maintenance

Cancer stem cells (CSCs) constitute a distinct minority subset of cells within tumors, characterized by stem cell-like features. It is imperative to clarify that CSCs and cancer-initiating cells are not synonymous; they represent different populations within the tumor microenvironment. CSCs are pivotal contributors to tumor initiation, progression, and resistance to therapy [23,40].

CSCs exhibit distinctive properties that significantly influence tumor behavior and therapeutic responses. One key characteristic, clonogenicity, underscores their ability to originate from a single cell and grow into a colony. Beyond clonogenicity, CSCs possess the unique capacity for self-renewal and differentiation, making them both tumorigenic and adaptable to various biological processes. This adaptability contributes to their substantial impact on tumor progression, therapeutic resistance, and disease recurrence. CSCs engage in sustained proliferation, invade normal tissue, promote angiogenesis, evade immune surveillance, and resist conventional anticancer therapies. The intricate regulation of CSC homeostasis involves critical transcription factors, including OCT4, SOX2, KLF4, NANOG, and c-MYC. Additionally, intracellular signaling pathways such as Wnt/TCF, STAT3, and NF-κB play pivotal roles in governing CSC phenotypes. The ATP-binding cassette (ABC) transporter family expressed through activation of NF-kB SREBP2, SNAIL, and TWIST transcription factors further contributes to the multi-drug resistance exhibited by CSCs [41]. To aid in the identification and isolation of these cells, cellular markers such as CD44, CD133, and ALDH1 are commonly employed tools. Altogether, these characteristics and regulatory mechanisms define the unique and influential role of CSCs in cancer biology [42].

In CSCs, autophagy appears to be upregulated and helps maintain their stem cell-like properties. Autophagy can promote the survival of CSCs by providing them with energy and nutrients during periods of stress or nutrient deprivation. Autophagy also helps CSCs to resist chemotherapy and radiation therapy by promoting DNA repair and reducing oxidative stress [43]. In this regard, Chen et al. suggested that the activation of autophagy is vital for the self-renewal of CSCs in head and neck squamous cell carcinoma (HNSCC) and the acquisition of CSC properties under adverse conditions. The results revealed that adverse conditions, such as exposure to cisplatin, starvation, and hypoxia, can increase the autophagy level and CSC properties of HNSCC cells. However, the CSC properties acquired under adverse conditions were reduced when pretreated with autophagy inhibitors, including 3-MA and chloroquine (CQ). The study also found that CSCs have stronger autophagic activity than non-CSCs, and the inhibition of autophagy dampens the CSC properties of HNSCC cells. Furthermore, the study identified the noncanonical FOXO3/SOX2 axis as the intrinsic regulatory mechanism that controls the CSC phenotype [44]. Also, Shi et al. indicated that the long noncoding RNA (lncRNA) TINCR regulates liver cancer stem cell (LCSC) self-renewal through autophagy activation via the polypyrimidine tract binding protein 1 (PTBP1) /ATG5 regulatory pathway. They found that TINCR is highly expressed in hepatocellular carcinoma (HCC) tissues and LCSCs, and its expression is crucial for the self-renewal and tumorigenesis of LCSCs. Gene ontology analysis uncovered that TINCR plays a role in maintaining stemness through the regulation of autophagy. Knocking down TINCR resulted in reduced expression of transcription factors POU5F1, SOX2, Nanog, and surface marker CD44. Mechanistically, TINCR was observed to interact with PTBP1 proteins, subsequently enhancing the transcriptional activity of the autophagy-related gene ATG5 [45].

Inconsistent with previous studies, other studies have shown an increase in the stemness properties of CSCs with a reduction in autophagy. In this regard, Brunel et al. explored the impact of silencing two autophagy-related genes (Beclin1 or ATG5) on the expression of markers and functionalities associated with glioblastoma CSCs. The results showed that downregulating autophagy increased the expression of CSCs markers and boosted cell proliferation and clonogenicity [46]. In addition, Park et al. revealed that luminal and triple-negative breast cancers (TNBCs) necessitate distinct treatment approaches due to variations in their autophagy flux levels. Their study uncovered the inhibition of autophagy flux in CSCs within TNBCs, coupled with the upregulation of miRNA-181a expression in both TNBC CSCs and patient tissues. ATG5 and ATG2B were identified as targets of miR-181a, playing a role in the early formation of autophagosomes [47].

Furthermore, Sharif et al. conducted a study demonstrating the role of autophagy in the TP73/p73-dependent regulation of stemness within CSCs. The study revealed that TP73/p73 positively regulates the growth and stemness of CSCs by modulating autophagy. Specifically, TP73/p73 deficiency promotes autophagy in CSCs by activating the autophagy machinery, resulting from reduced ATP levels caused by metabolic perturbations within the proline regulatory axis. These findings suggest that autophagy may contribute to a decrease in stemness maintenance in CSCs [48].

### 2.3. Autophagy as a Therapeutic Target in Cancer

Autophagy is believed to be a double-edged sword in cancer, exhibiting the potential to either promote tumor cell survival or induce cell death, contingent on the specific context, and its manipulation by drugs needs to be carefully considered to achieve therapeutic benefits [49]. In this regard, a growing body of experimental studies and clinical trials have attempted to demonstrate the efficacy of targeting autophagy as a therapeutic approach to overcome cancer. For instance, Yun et al. suggested that autophagy, induced by temozolomide (TMZ) treatment, may contribute to resistance in glioblastoma multiforme (GBM). They discovered that DOC-2/DAB2 interacting protein (DAB2IP) loss causes TMZ resistance in GBM via ATG9B, while DAB2IP sensitizes GBM to TMZ leading to autophagy suppression through ATG9B expression regulation. In comparison to low-grade glioma, GBM cells showed higher ATG9B expression, and enhanced knocking down of ATG9B, which in turn decreased autophagy and resistance to TMZ. Furthermore, the study demonstrated that DAB2IP negatively regulates ATG9B expression by blocking the Wnt/β-catenin pathway. To prevent therapeutic resistance and enhance the benefit of TMZ, the study tested effective combination strategies using a small molecule inhibitor blocking the Wnt/β-catenin pathway along with TMZ. The combined treatment showed a synergistic enhancement of TMZ efficacy in GBM cells, providing insight into a potential strategy to overcome TMZ resistance [50]. Moreover, Chen et al. investigated the role of autophagy in the treatment of HCC and demonstrated that the combination of hydroxychloroquine (HCQ) as an autophagy inhibitory agent and sorafenib had a synergistic effect on sorafenib-resistant HCC cells. This was achieved by downregulation of Toll-like receptor (TLR)-9. TLR-9 increased HCC cells proliferation, tumor growth, oxidative markers, and autophagosome formation. Therefore, TLR-9 modulation resulted in a decrease in autophagy-related genes (ATG5 and Beclin-1), oxidative stress markers (SOD1), apoptosis-related genes (c-caspase3), and finally, tumor growth. These findings suggest that targeting autophagy with HCQ and downregulating TLR9 could increase the sensitivity of sorafenib-resistant HCC cells to treatment. The modulation of autophagy may therefore be a promising strategy in cancer therapy [51]. In addition, Li et al. investigated the role of autophagy in osimertinib resistance in non-small cell lung cancer (NSCLC) and found that increased autophagy was associated with resistance to the third-generation epidermal growth factor receptor-tyrosine kinase inhibitor (EGFR-TKI). Suppression of autophagy improved osimertinib cytotoxicity in both resistant and sensitive NSCLC cells, indicating a detrimental role of autophagy in osimertinib efficacy. These findings highlight the potential of combination therapy with an EGFR-TKI and an autophagy inhibitor as a promising strategy to overcome osimertinib resistance in lung cancer [52]. Moreover, You et al. suggested a novel relationship between breast cancer 1 (BRCA1) and autophagy in drug resistance. They investigated the role of BRCA1 and autophagy in the drug resistance of epithelial ovarian cancer stem cells (EOCSCs). Autophagy played a crucial role in preserving stemness and conferring resistance to chemotherapy in EOCSCs. BRCA1 was identified as a regulator of drug resistance in EOCSCs through autophagy, and inhibition of autophagy activity effectively reduced resistance caused by BRCA1. Additionally, BRCA1 regulated cellular apoptosis and cell cycle progression through autophagy, influencing drug sensitivity indirectly in EOCSCs [53].

Targeting autophagy to treat cancer was evaluated in several clinical trials. Karacis et al. in a phase 2 randomized clinical trial investigated whether the autophagy inhibitor, HCQ, in combination with standard chemotherapy improved overall survival at 1 year among patients with metastatic pancreatic cancer. The trial enrolled 112 patients, with 55 receiving gemcitabine hydrochloride and nab-paclitaxel (GA) plus HCQ and 57 receiving GA alone. The primary endpoint, overall survival at 12 months, was not improved by the addition of HCQ, although there was an improvement in the overall response rate. The study suggests that routine use of GA plus HCQ for metastatic pancreatic cancer without a biomarker is not supported, but HCQ may have a role in the locally advanced setting [54]. In another randomized phase 2 study, the efficacy of adding HCQ to preoperative chemotherapy was assessed in patients with pancreatic adenocarcinoma. Patients were randomly assigned to receive either GA alone or with HCQ for two cycles before resection. The study found that the addition of HCQ resulted in greater histopathologic and biomarker responses, as well as evidence of autophagy inhibition and immune activity. However, overall and relapse-free survival did not differ between the two arms. The study concludes that adding HCQ to chemotherapy may have potential clinical benefits in resectable pancreatic adenocarcinoma [55]. In another phase 2 clinical trial, Arora et al. compared the efficacy and safety of the histone deacetylase inhibitor, vorinostat, and HCQ to the multi-kinase inhibitor, regorafenib, in patients with metastatic colorectal cancer [56]. The study found that the median progression-free survival was inferior in the vorinostat/ HCQ arm compared to the regorafenib arm, although both treatments were well-tolerated and showed improved anti-tumor immunity. The study concludes that autophagy modulation using vorinostat/HCQ has a favorable safety profile and may have clinical benefits in metastatic colorectal cancer. Additionally, Xu et al. conducted a meta-analysis to evaluate the clinical value of autophagy inhibitor-based therapy in cancer treatment. They identified seven clinical trials and found that autophagy inhibitor-based therapy had higher overall response rate, progression-free survival rate, and overall survival rate compared to therapy without inhibiting autophagy. The study suggests that inhibiting autophagy may be a promising strategy for cancer treatment [57].

## 3. Cell Transformation, EMT and Carcinogenesis

EMT is a partially reversible cell plasticity program that occurs under various physiological and pathological conditions, playing a vital role in processes such as growth and development, wound healing, tissue fibrosis, and tumorigenesis [58]. The EMT process involves the alteration of normally arranged stationary compact epithelial cells into elongated motile mesenchymal cells under the influence of EMT-inducing signals [59]. During tumor development this process is associated with various outcomes, including tissue invasion, metastasis, resistance to cancer treatment, poor survival, and an increased risk of tumor recurrence [60]. The activation of EMT has been found to be essential for malignant progression in several types of cancers derived from epithelia, such as breast cancer, colorectal cancer, HCC, lung cancer, prostate cancer, and pancreatic cancer [17,61,62].

### 3.1. Morphological, Biochemical and Molecular Changes during EMT

Epithelial and mesenchymal cells differ in their morphology, epithelial/mesenchymal marker expression, tissue organization, and biological functions [63]. In normal epithelial tissues, flat and polygonal epithelial cells form monolayer or multilayer continuous sheets, serving as vital barriers for solutes and water [64]. Crucially, epithelial cells exhibit strong apical-basal cell polarity, which plays a fundamental role in preserving cellular architecture and facilitating essential biological functions like endocytosis, exocytosis, and vesicle transport across the epithelium [64,65]. Various cellular junctions, such as tight junctions, cadherin-based adherent junctions, gap junctions, desmosomes, and hemi-desmosomes, play a crucial role in maintaining the cohesion of epithelial cells and their connection to the underlying basement membrane [66,67]. Such cell–cell junctions are essential for preserving cell polarity, tissue structural integrity, and upholding their barrier function [68,69]. The key component of adherent junctions, located in the basolateral membrane and encoded by the CDH1 gene, is Epithelial-cadherin (E-cadherin). The extracellular domain of E-cadherin binds to adjacent epithelial cell cadherin, while its intracellular domain interacts with β-catenin, α-catenin, and p120-catenin, forming the cadherin/catenin adhesion complex. This complex participates in intracellular signaling as well as anchors to cortical actin bundles [69]. On the other hand, desmosomes are associated with cytokeratin intermediate filaments [70]. In addition to tight junctions, several multi-protein polarity complexes play a role in regulating epithelial apico-basal polarization. These complexes include Partitioning defective (PAR), Crumbs, and Scribble complexes, which exhibit a cortically asymmetric distribution [62,67,71]. The mobility of individual cells within the epithelial layer are restricted due to their epithelial morphology and their cellular junctions [63,72].

Unlike epithelial cells, mesenchymal cells exhibit distinct features such as irregular morphology, am elongated spindle-like shape, less-rigid topography, anterior–posterior polarity, and the presence of vimentin-based intermediate filaments instead of epithelial cytokeratins [65,67]. Functional epithelial cell junctions are not observed in mesenchymal cells, thereby resulting in a limited ability of these cells to form strong connections with surrounding mesenchymal cells [63]. In mesenchymal cells, several key factors contribute to their ability to move and migrate. Firstly, there is a downregulation of cytokeratin and an upregulation of vimentin filaments, which enhances the strength and flexibility of the cytoskeleton and reduces its susceptibility to damage during migration [73,74,75]. Additionally, mesenchymal cells produce actin-rich membrane protrusions through N-cadherin-mediated stress fiber organization which facilitate various types of cell movement [74,76,77]. Moreover, mesenchymal cells gain expression of α-smooth muscle actin (α-SMA), which favors the contractility of the extracellular matrix (ECM) proteins in which cells are embedded [78].

The transition between epithelial and mesenchymal cell states, referred to as the EMT process, has been observed in diverse cellular fate conversions, including cancer progression [79]. EMT is associated with profound cellular changes at multiple levels, encompassing dynamic morphological and biochemical alterations that result in functional modifications in cell migration and invasion [62,67]. These changes include extensive cytoskeletal remodeling and alterations in their proteins expression such as a switch from cytokeratins to vimentin, the formation of actin stress fibers, enhanced production of ECM-degrading enzymes, alterations in the expression of certain cell surface proteins, disintegration of epithelial cell junctions, loss of apical-basal polarity, transcriptional repression of polarity complex proteins, and the acquisition of a fibroblastoid invasive phenotype [62]. EMT is often not a totally defined irreversible process. Indeed, during tumor transformation terms such as partial EMT (pEMT), hybrid epithelial/mesenchymal (E/M), and mixed EMT have been used, suggesting that cells range across a continuum between the full epithelial towards the mesenchymal state [80]. Moreover, cellular and molecular markers that reflect the characteristics of epithelial and mesenchymal cells, including specific parameters involved in cell integrity and regulatory processes, can be used as markers for EMT [67].

The downregulation or loss of E-cadherin, a major hallmark of EMT, results in the disintegration of adherence junctions. It also leads to the dissociation of membrane-bound β-catenin from the cadherin complex and its translocation to the cell’s nucleus. Within the nucleus, β-catenin regulates the expression of various genes, including c-myc, cyclin D1, and genes associated with Wnt signaling [61,63,81]. Cadherin isoform switching, characterized by a change in the expression pattern of cadherin from E-cadherin to N-cadherin, is an important phenomenon in EMT. Such changes significantly affect the motility of cells undergoing EMT and lead to an increase in their ability to interact with stromal cells [82,83]. Cytokeratins (CK8, CK18, CK19), mucin-1, desmoplakin, and tight junction proteins such as occludin, claudin, and zone of occlusion-1 (ZO-1) are also important markers which are downregulated in epithelial cells undergoing EMT. Under these circumstances, there is an upregulation of mesenchymal cell-specific markers including vimentin, α-SMA, fibroblast-specific protein-1 (FSP-1), fibronectin, vitronectin, β1 and β3 integrins, and mesenchymal cadherins such as N-cadherin [60,63,84].

Being that EMT induces a global reprogramming of the cell proteome, several EMT transcription factors (EMT-TFs), epigenetic modifications, post-translational alterations, and specific non-coding RNAs (ncRNAs) including lncRNAs, as well as microRNAs like the miR-200 family, play a role in regulating specific cellular processes [63,85,86]. The key EMT-TFs include the basic helix-loop-helix (bHLH) TFs Twist1 and Twist2, the Snail family of zinc-finger TFs, Snail (SNAI1) and Slug (SNAI2), and the zinc finger E-box binding homeobox (ZEB) family, specifically ZEB1 and ZEB2 [87]. Upon activation of specific signaling cascades, EMT-TFs have the ability to repress the transcription of epithelial markers. In particular, EMT-TFs can directly inhibit E-cadherin expression by binding to E-boxes located on the CDH1 promoter [85,88]. In addition to the extensively studied EMT-TFs, several non-canonical EMT-TFs have been identified, which may play a role in coordinating the EMT process in specific cancers. These include Krüppel-like factor 8 (KLF8), placenta-related homeobox1 (PRRX1), and fork boxC2 (FOXC2) [85]. Several signaling pathways contribute to the induction of the EMT process. These pathways include transforming growth factor-β (TGF-β), bone morphogenetic protein (BMP), Wnt-β-catenin, NOTCH, sonic hedgehog (Shh), nuclear factor kappa B (NF-κB), AKT-mTOR, integrins, and receptor tyrosine kinases (RTKs) including epidermal growth factor (EGF) and fibroblast growth factor (FGF) [89]. Among these factors, TGF-β is recognized as the most potent inducer of EMT, exerting its effects through both Smad and non-Smad pathways, ultimately leading to the upregulation of EMT-TFs [90,91].

### 3.2. Role of EMT in Tumor Invasion and Metastasis

The processes of EMT and its reversal, known as mesenchymal-to-epithelial transition (MET), have been identified as critical events in the metastatic cascade of epithelial cancers [61,92].

Tumor cell invasion, which refers to the directed migration of tumor cells from the tumor microenvironment into surrounding tissues by crossing the basement membrane, is considered the initial stage in the metastatic cascade [93]. The ability of tumor cells to invade neighbor cells and tissues is crucial for metastasis initiation. However, subsequent steps such as intravasation, transport, extravasation, and colonization at secondary sites are necessary to drive the metastatic cascade [94].

Cell invasion occurs by two main strategies, individual migration and collective migration [27,68]. In single-cell invasion, cell adhesion to ECM components plays a crucial role. However, in collective invasion, coordinated cell–cell interactions are essential [95]. Collective invasion employs a migration mechanism known as collective–amoeboid transition, wherein cells acquire an amoeboid phenotype [93,96]. During tumor invasion, single tumor cells can exhibit two distinct modes of movement: mesenchymal and amoeboid migration, which can be interchangeable [97]. Amoeboid migration is characterized by membrane blebbing, weak adhesion or pushing movements, high myosin II activation, and rapid motility. On the other hand, mesenchymal migration involves ECM degradation, strong adhesions, front-rear polarization, and the formation of actin-rich membrane protrusions like lamellipodia and filopodia. Indeed, epithelial cells can adopt a migratory phenotype by converting to a mesenchymal state through the activation of EMT. These mesenchymal-like cells may be able to increase the speed of migration through complex three-dimensional environments by acquiring amoeboid characteristics, referred to as mesenchymal-to-amoeboid transition (MAT) [53,95,98,99]. In certain epithelial cancers, such as HCC, prostate cancer, and breast cancer, direct epithelial-to-amoeboid transition (EAT) has been observed as an additional migration pattern that enhances tumor cell invasion and metastasis [95].

The aberrant activation of the EMT process is considered a significant event in initiating the invasion and dissemination of cancer cells [100]. Additionally, partial EMT has been observed in the context of metastasis. During this state, cells are capable of collective migration and demonstrate an enhanced metastatic potential [9]. Multiple studies conducted on various cancer models, including tumor cell lines, mouse models of cancer, and human tumor samples, have confirmed that activating EMT is a crucial strategy for facilitating the effective metastatic spread of tumor cells [101]. These studies reveal that epithelial cancer cells can acquire a mesenchymal phenotype and express mesenchymal markers such as α-SMA, FSP1, and vimentin. Typically, these cells are localized at the invasive front of primary tumors and eventually undergo subsequent stages of the invasion–metastasis cascade [102].

Evidence shows a connection between the process of EMT and protein hydrolases, specifically matrix metalloproteinases (MMPs). At the invasion front, individual cells or cell clusters possessing EMT properties can invade and disrupt the ECM by activating proteases like MMPs [64]. A significant association has been identified between tumor aggressiveness and increased expression of MMPs, particularly MMP-1, -2, and -9 in breast, lung, pancreatic, and prostate cancers [60,103,104]. ECM degradation by certain classes of MMPs is considered a prerequisite for cell invasion [105]. Tumor cells undergoing EMT can produce higher levels of MMP enzymes to enhance cell invasion and migration [60]. EMT-TFs have the ability to induce MMPs, which play a crucial role in degrading the basement membrane and ECM of surrounding tissues [62]. Previous studies have demonstrated that Twist and ZEB1 can stimulate the formation of invadopodia [62,106]. Invadopodia are specialized filopodia formed by invasive cancer cells, enabling them to degrade the ECM through MMP secretion [107]. Additionally, an association has been observed between increased MMP levels in the tumor microenvironment and the activation of EMT in epithelial cells. Various types of epithelial cells, including those in ovarian, lung, prostate, lens, and breast tissues, have exhibited MMP-dependent induction of EMT [64,108]. MMP-mediated degradation of E-cadherin, as an important substrate for MMPs, leads to tissue conversion into single cells and triggers EMT signaling. The extracellular domain of E-cadherin undergoes proteolytic cleavage, giving rise to an 80 kDa fragment identified as soluble E-cadherin (sE-cad), which is believed to be involved in functions such as inducing EMT and facilitating invasion [109].

EMT is a partly reversible process characterized by the transition from motile, multipolar, or spindle-shaped mesenchymal cells to planar arrays of polarized cells organized in epithelia. Once tumor cells reach suitable metastatic sites, complete or partial EMT may be followed by the MET process. MET involves the reorganization of the cytoskeleton, restoration of epithelial cell-to-cell junctions, and establishment of apical-basal polarity (Figure 2). These changes enable migrating mesenchymal cells to adopt a polarized epithelial state, facilitating colonization at secondary sites [61,63,67]. For instance, a study utilizing a mouse model of skin cancer demonstrated that the activation of Twist1 induces EMT and facilitates the dissemination of cancer cells through the bloodstream. Conversely, in distant tissues, the downregulation of Twist1, which triggers the MET process, is necessary for the proliferation of disseminated cancer cells and the formation of macrometastases [110]. Thus, the dynamic nature of EMT-MET cellular processes can lead to alterations in the phenotype of tumor cells in a spatiotemporal manner [62].

### 3.3. EMT as a Target in Cancer Therapy and Drug Resistance

In addition to the promotion of increased cell motility and invasiveness described above, EMT has been demonstrated to play a role in other aspects of tumorigenesis, including drug resistance or chemoresistance [62,111]. Since 1990, numerous studies using different models (in vitro, in vivo, and clinical samples) have investigated the relationship between EMT and drug resistance. There are several lines of studies in support of association of EMT and drug resistance of cancer cells and underlying mechanisms in various cancers, including lung, pancreatic, bladder, and breast cancers [112,113].

EMT-related TFs, such as Twist, Snail, Slug, and ZEB, play crucial roles in promoting drug resistance in cancer cells. Several binding sites for these TFs have been identified in the promoters of the ATP-binding cassette (ABC) transporter family, which includes multidrug resistance (MDR) proteins. These transmembrane proteins actively participate in drug efflux [114,115,116]. For instance, in invasive breast cancer cell lines, overexpression of EMT-related TFs including Twist, Snail, and FOXC2 results in an increase in the promoter activity and expression of ABC transporters, thereby enhancing resistance to doxorubicin treatment [117]. Similarly, overexpression of Twist in human colorectal cancer cell lines induces EMT and heightens resistance to oxaliplatin treatment by upregulating MDR1 [116].

Another well-known mechanism contributing to drug resistance in EMT is the inhibition of drug-induced apoptosis signaling pathways [118]. For instance, a study demonstrated that the expression of Slug, a member of the Snail TF superfamily, is significantly higher in cancer cells with resistance to EGFR tyrosine kinase inhibitors and can promote gefitinib resistance in NSCLC by suppressing Bim expression and inhibiting caspase-9 activity. This finding was supported by showing that silencing this TF restored the gefitinib-mediated apoptosis [119].

EMT-inducing signaling pathways such as TGF-β, Wnt, and Hedgehog have also been implicated in drug resistance [118]. In this connection, treatment of HCT116 human colon cancer cells with doxorubicin resulted in EMT induction through the activation of TGF-β signaling and increased MDR levels. Furthermore, by suppression in TGF-β signaling through downregulating Smad4, the EMT process was reversed and the sensitivity of the cells to doxorubicin was restored [120].

Cells undergoing EMT may exhibit similar characteristics to CSCs in terms of marker expression, involvement of key signaling pathways such as Wnt, Hedgehog, and Notch, as well as the development of a drug-resistant phenotype [118,121]. Wnt and Hedgehog, for instance, may favor the expression of CSC markers such as CD34, CD44, Sca, and cKIT [122,123]. Indeed, CSCs show resistance to several chemotherapy drugs due to overexpression of drug efflux transporters, such as ABC transporters, slow-cycling (dormant) nature, prevention of apoptosis, and expression of non-coding RNAs involved in drug resistance [124].

Factors associated with the tumor microenvironment, including hypoxic conditions, also play an important role in promoting EMT-related chemoresistance [118]. Under hypoxic conditions, the activation of hypoxia-inducible factor 1-alpha (HIF-1α) can induce EMT in HCC cells and enhance drug resistance by upregulating MDR1 expression, suggesting that HIF-1α suppression can enhance the efficacy of chemotherapy [125].

Relationships between the EMT process and resistance to other cancer treatments such as radiotherapy, molecular targeted therapy, and immunotherapy further confirm the role of EMT in drug resistance [126]. Based on this data, various therapeutic strategies have been proposed, including targeting EMT-TFs and inhibitors of signaling pathways involved in the EMT process, such as TGF-β/Smad signaling. Moreover, several small molecules and microRNAs have been identified as EMT inhibitors that are able to reverse the EMT phenotype and reduce tumor resistance to therapy. Long-term use of the EMT inhibitors such as disulfiram and metformin raises concerns over the side effects and toxicity, as well as the role of the MET process in cancer metastasis [118,126]. It is important to note that intratumor heterogeneity and the plasticity of cancer cells during the EMT-MET process can give rise to different resistance mechanisms [127,128]. According to Biddle et al., depending on cellular states and plasticity, the EMT inhibitors can be classified into different sub-groups [127].

These data indicate that EMT process and related factors can be used as target for controlling cancer cells at early stages of tumorigenesis. However, targeting these cells and molecules depends on different factors, particularly when used in combination with conventional chemotherapy, immunotherapy, or other cancer treatment protocols.

## 4. Autophagy-EMT Interplay

As described in the previous section, autophagy is a highly conserved cellular pathway that facilitates the elimination of harmful or defective biomolecules and organelles. It can either stimulate or repress tumor formation depending on the type of tissue, tumor stage, metabolic context, and microenvironmental inputs [129]. The EMT plays a key role in tumor invasion and metastasis by enabling epithelial cells to acquire mesenchymal properties, such as motility and the ability to spread and disseminate in distant organs [91] as well as being implicated in the acquisition of chemoresistance [130]. Interestingly, autophagy and EMT share a complex regulatory network and several stress-related signaling pathways. Therefore, understanding their interrelations may open up novel therapeutic opportunities in cancer treatment.

The dual role of autophagy during tumor development has been described in previous sections. The use of autophagy as a cancer therapeutic target is highly debated. On one hand, autophagy can be induced in cancer in response to chemotherapy damage, and several pharmacological studies are underway to prevent autophagy, mainly by using CQ or HCQ. On the other hand, further investigations are currently ongoing to inhibit mTOR signaling, which can activate the tumor suppressor mechanism of autophagy [131].

EMT governs the most lethal aspects of cancer and is therefore an attractive target for cancer therapy. Although specific targeting of EMT-related molecules in clinics is challenging, recent studies have identified various metabolic and autophagy-related pathways involved in EMT. It has been proposed that using drugs that are FDA approved or currently tested in clinical trials and are active in metabolism and/or autophagy could represent a valid repurposing strategy to target EMT in cancer [132]. Metabolism-inhibiting drugs or autophagy modulators may be used with standard chemo- or immunotherapy to treat EMT-driven resistant and aggressive cancers.

Recent observations have unveiled complex interactions between autophagy and EMT. Previous investigations have revealed that EMT-related molecules and signaling pathways may activate autophagy. Several extracellular signals that induce EMT also activate autophagy, such as the Wnt/β-catenin, TGF-β, HIF-1α, and Notch signaling pathways [133]. Intriguingly, autophagy is implicated in the regulation of EMT, mainly through the activation of energy response pathways, triggering EMT-inducing signaling pathways or controlling the degradation of EMT-related adhesion and cytoskeletal molecules as well as EMT-TFs [134,135,136,137]. Due to its multifaceted function in tumor formation and development, autophagy’s impact on EMT is debatable and highly dependent on the type of cell and tissue, stage of tumor, and the metabolic and microenvironmental inputs that modulate this process [138].

Autophagy has been shown to promote EMT in certain contexts. For example, autophagy has been reported to promote EMT in breast cancer cells by activating the TGF-β/Smad signaling pathway [90,135]. Autophagy can act as a pro-tumor process by providing the highly invasive and metastatic cancer cells with their energy needs [133,139]. In this regard, it has been demonstrated that an EMT-like phenotype occurs alongside a greater autophagy flux [140]. The pro-survival function of autophagy against cell apoptosis cues resulting from alterations to adhesion and cytoskeleton remodeling may be the driving factor for the metastatic process [138]. Thus, inhibiting autophagy in several types of cultured cancer and non-cancerous cells may block EMT [141]. An enhancement in EMT inhibition has been demonstrated in renal cancer cells when combining the autophagy blocker CQ and the typical chemotherapy treatment [140]. In another study, it has been reported that CQ reverses paclitaxel resistance and reduces metastatic potential in A549 lung cancer cells via ROS-mediated regulation of the β-catenin pathway [142].

Conversely, a number of studies suggest that autophagy may also interfere with metastatic dissemination, acting as a tumor suppressor mechanism, and that cancer cells may be prevented from acquiring an EMT-like phenotype by activating autophagy [143,144,145]. Recent studies suggest that autophagy induction by mTOR inhibitors, bioactive molecules, or environmental stresses could revert the EMT phenotype in several in vitro and in vivo models [141,146,147,148]. A study by Catalano et al. demonstrated a molecular shift from a mesenchymal to an epithelial-like phenotype in GBM cell lines by overstimulation of autophagy following energy starvation or mTOR inhibition. This phenotypic change drastically affected migration and chemokine-mediated invasion, providing a further rationale for including autophagy modulators in the current therapeutic regimen of GBM patients. At the molecular level, a down-regulation of key EMT-TFs, such as Slug and Snail, was observed upon autophagy induction, and consequently, a transcriptional and translational up-regulation of N- and R-Cadherins expressed in the neural tissue [149].

In addition to its role in regulating EMT, autophagy has also been implicated in the maintenance of stemness in cancer cells. CSCs have been shown to be in an EMT-like state. Autophagy has been reported to promote the maintenance of stemness in CSCs by suppressing some EMT-like features. In breast cancer cells, the inhibition of autophagy leads to the induction of EMT and the loss of stem-like properties [43,150]. On the other hand, the regulatory role of the EMT process on autophagy has been revealed through inducing metabolic alterations or modulating the expression of autophagy-related genes mediated by EMT-TFs [151,152,153].

In the following sections, we delve into the complex regulatory mechanisms and the interplay of signaling pathways regulating autophagy and EMT in cancer, as well as their significance during tumor progression and metastasis. We also explore how the degradation of EMT-related markers by the autophagolysosomal system affects phenotypic changes during EMT and its repercussions in cancer metastasis.

### 4.1. Common Signaling Pathways That Regulate Both Autophagy and EMT

Recent studies have revealed a complex interplay between signaling pathways regulating EMT and autophagy. The EMT process is regulated by various molecular and cellular signaling pathways, including Wnt, TGF-β, and Notch signaling, which have also been shown to regulate autophagy.

One key pathway that has been implicated in the interplay between EMT and autophagy is the Wnt/β-catenin signaling pathway [154]. Wnt signaling is a critical regulator of EMT and has been shown to promote the mesenchymal phenotype in cancer cells. In addition, Wnt signaling has also been shown to activate autophagy through the inhibition of mTOR signaling [141]. This suggests that the activation of Wnt signaling during EMT may promote both the mesenchymal phenotype and autophagy, which could have important implications for cancer progression. Certain growth factors, which are responsible for triggering EMT-specific TFs [155,156], are additional EMT-inducing cues that originate in the tumor microenvironment. It has been demonstrated that the TGF-β signaling pathway promotes the mesenchymal phenotype in cancer cells and is a crucial regulator of EMT. To put it another way, TGF-β signaling has been shown to promote autophagy while also inducing EMT [157]. TGF-β activates the AMPK pathway and inhibits the mTOR pathway, which together induce autophagy [158]. However, the role of EGF and platelet-derived growth factor (PDGF) in autophagy has also been described. In melanoma models and vascular smooth muscle cells, they can lead to ERK1/2- and mitogen-activated protein kinase )MAPK(-dependent autophagy pathways upon starvation [159,160].

Another important pathway involved in the interplay between EMT and autophagy is the nuclear factor erythroid 2-related factor 2 )NRF2(/Kelch-like ECH-associated protein1 )Keap1( pathway [161]. This pathway is important for controlling how cells react to oxidative stress and is connected to both EMT and autophagy. The activation of NRF2 signaling during EMT has been shown to promote the mesenchymal phenotype in cancer cells, and may also promote autophagy through the upregulation of autophagy-related genes [162]. Conversely, the inhibition of NRF2 signaling has been shown to promote the reversal of EMT and the inhibition of autophagy in cancer cells [163].

Tumor Necrosis Factor-α (TNFα), which may facilitate EMT induction via the NF-ҡB signaling pathway stimulation [164], leads to the expression of the proinflammatory cytokine interleukin 6 (IL6) gene, which, in turn, activates autophagy in melanoma cells [165]. It has been found that NF-κB can directly regulate the transcription of EMT-TF genes in human breast cancer cell lines [166]. NF-κB also may modulate autophagy. By directly stimulating the expression of genes or proteins involved in the autophagosome such as BECN1, ATG5, and LC3, IKK/NF-κB signaling may cause autophagy [167]. However, in tumors such as Ewing sarcoma, breast, and promyelocytic leukemia, NF-κB induced by TNF-α may repress autophagy via activation of mTOR pathway [168,169].

HIF-1α, which induces EMT and a metastatic phenotype through the regulation of ZEB1 and Snail in colorectal and gastric cancer cells [170,171], also plays a key role in autophagy regulation through the p27-E2F1 signaling pathway and ATG regulation under hypoxic conditions [172,173].

Moreover, the Beclin-1 signaling pathway has been implicated in the interplay between autophagy and EMT in cancer [174]. In fact, Beclin-1 is a key regulator of autophagy, and functions as a part of the PI3K complex that is required for the initiation of autophagy [175]. Beclin-1 interacts with other proteins, including Bcl-2, to regulate the activation of autophagy. Several studies have suggested that Beclin-1 may also play a role in the regulation of EMT in cancer. For example, one study found that the expression of Beclin-1 was decreased in breast and thyroid cancer cells undergoing EMT, and that the overexpression of Beclin-1 inhibited the EMT process [176,177]. This suggests that Beclin-1 may act as a negative regulator of EMT in cancer cells. Conversely, other studies have suggested that the upregulation of Beclin-1 may promote the mesenchymal phenotype in cancer cells. For example, one study found that the overexpression of Beclin-1 induced EMT in HCC cells, and the PI3K/AKT/mTOR pathway was activated to mediate this process [178,179].

In addition to these pathways, several other molecular mechanisms have been implicated in the interplay between EMT and autophagy. These include the Hippo signaling pathway, which has been shown to regulate both EMT and autophagy in cancer cells, and the Notch signaling pathway, which has been implicated in the regulation of both EMT and autophagy in a variety of cell types [133,180,181]. Based on evidence, IL-6/STAT3 (signal transducer and activator of transcription-3) signaling and integrin-focal adhesion signaling are other known signaling pathways involved in the regulation of both EMT and autophagy processes in cancer [139].

The interplay between EMT and autophagy has also been shown to play a role in regulating the immune response, particularly in the context of cancer. In fact, EMT has been shown to contribute to immune evasion by cancer cells [182]. During EMT, cancer cells downregulate the expression of major histocompatibility complex (MHC) class I molecules that are recognized by the immune system [183,184]. The downregulation of cell surface molecules such as E-cadherin can help cancer cells evade immune recognition and attack. Autophagy, on the other hand, has been shown to play a role in regulating the immune response by modulating the presentation of antigens by MHC class II molecules. Autophagy can promote the degradation of intracellular pathogens and the generation of antigenic peptides that can be presented by MHC class II molecules to activate CD4^+^T cells, a critical component of the immune response [185]. Moreover, autophagy has been shown to modulate the activation of dendritic cells, which are key regulators of the immune response. In summary, the interplay between EMT and autophagy is involved in the regulation of the immune response, particularly in the context of cancer. In order to design strategies for manipulating EMT and autophagy to improve the immune response to cancer, more study is required to completely understand the molecular mechanisms behind this interplay. Figure 3 indicates an overview of signaling pathways that regulate the interplay between autophagy and EMT.

### 4.2. Autophagy Degradation of EMT-Related Markers

Autophagolysosomes and chaperone-mediated autophagy degradation of EMT-related molecules has been investigated, revealing the presence of a complex oncogenic interplay highly regulated between EMT and autophagy [136,186]. A number of studies highlight that the selective degradation of key EMT-associated factors such as epithelial cell adhesion proteins represents one of the main ways by which autophagy might influence EMT [187]. A schematic demonstration of different potential interplays between EMT and autophagy is depicted in Figure 4.

Autophagy can promote EMT by degrading E-cadherin, a key epithelial marker [188]. The loss of epithelial polarity and the development of a mesenchymal phenotype are favored by the degradation of E-cadherin by autophagy. Damiano et al. found that autophagy decreases the E-cadherin protein levels by transporting it to the autophagosome via the autophagic cargo adaptor SQSTM1/p62 [189]. According to Zhou et al., plant homeodomain finger protein 8 (PHF8) has an oncogenic function in HCC by accelerating FIP200-dependent autophagic degradation of E-cadherin, which leads to EMT and metastasis [190]. This study sheds light on the significance of E-cadherin autophagic degradation in HCC and suggests PHF8 as a novel viable target for HCC therapy. In another study, it has been reported that E-cadherin protein expression was decreased upon TGF-β1 treatment in mouse kidney proximal tubular epithelial cells concomitantly with the increase in LC3-II, the autophagy marker. Interestingly, the use of inhibitors of the autophagy–lysosomal pathway, including ammonium chloride or CQ, was able to rescue TGF-β1-mediated E-cadherin degradation. In contrast, the treatment of the proteasome inhibitor MG132 was not able to prevent the TGF-β1-induced decrease in E-cadherin [191]. These findings point to a role for autophagy in E-cadherin degradation in TGF-β1-induced EMT. The NAD-dependent deacetylase sirtuin-1 (SIRT1) was found to induce EMT by promoting E-cadherin degradation via autophagy and facilitating melanoma metastasis. SIRT1 boosted the autophagic breakdown of E-cadherin by deacetylating Beclin-1. Furthermore, autophagy inhibition could rescue E-cadherin protein levels and repressed cell migration and invasion by preventing the degradation of E-cadherin in SIRT1-overexpressing cells [192].

Differently from SIRT1, inhibition of deacetylase 1 (HDAC1) by MS-275 was demonstrated to promote EMT reversal and at the same time to induce autophagy [193,194].

According to Xu et al., E-cadherin stability could also be influenced by tripartite motif-containing 29 (TRIM29), which has been shown to lead its autophagic degradation through Beclin-1 in HTB-182 and NCL-H1915 lung squamous carcinoma cell lines. E-cadherin degradation by TRIM29 could finally boost the EMT program, which enhances migration and invasion in cellular models [195].

Although autophagic degradation of E-cadherin may enhance the activation of the EMT program, other studies have shown the role of autophagy in regulating the degradation of EMT inducers, such as mediators of EMT-related signaling pathways, ECM, and cytoskeleton-related proteins, as well as EMT-TFs, which may limit the metastatic potential of cancer cells.

The Wnt/β-catenin signaling pathway is frequently up-regulated in colorectal tumors [196,197]. β-catenin forms complexes with the TCF/LEF family of TFs to regulate target gene expression [198]. A number of studies attribute a key role to the Wnt/β-catenin pathway for the induction of the EMT program in different models and upon several stress conditions [199,200]. During nutrient deprivation, β-catenin can be selectively targeted for autophagy clearance following the formation of a β-catenin–LC3 complex which impedes the β-catenin /TCF-driven transcription required for metabolic stress adaptation [201,202]. In GBM cells, autophagy has been reported to negatively regulate the Wnt/β-catenin signaling pathway by promoting β-catenin delocalization intracellularly, mainly to submembrane regions which, in turn, limits the nuclear translocation of β-catenin [203]. On the other hand, Wnt/β-catenin may also repress autophagy and p62 expression, suggesting the existence of a regulatory feedback mechanism [201].

The ECM glycoprotein fibronectin1 (FN1) is known as a key marker of EMT and metastasis, which can facilitate tumorigenesis [204,205]. It has been demonstrated in HNSCC that upregulation of FN1 is correlated with poor prognosis and a high tumor grade [206]. In a human oral squamous carcinoma cell line (SCC-25), it was shown that inducers of autophagy, such as rapamycin and Earl’s balanced salt solution (EBSS), lead to increased degradation of FN1. Conversely, the use of autophagy inhibitors, including bafilomycin A1 (Baf A1), 3-MA and CQ, leads to reduced degradation of FN1. X Liu et al. reported the mechanism of p62/SQSTM1-mediated autophagy–lysosomal degradation of FN1 in HNSCC [188].

Vimentin is a mesenchymal marker involved in cytoskeletal organization and focal adhesion turnover and strongly correlates to tumorigenesis and EMT induction [207,208]. The autophagy adapter protein SQSTM1/p62 has been shown to be overexpressed in breast cancer, and this protein is considered as a metastasis-related protein [209]. P62/SQSTM1 has been reported to bind to and stabilize vimentin, which in turn promotes cancer cell invasion and metastasis. Notably, depletion of p62/SQSTM1 can downregulate vimentin protein expression independent of the ubiquitin–proteasome pathway [210], suggesting a possible role for the autophagy–lysosomal pathway in vimentin degradation.

### 4.3. Reciprocal Regulation between Autophagy and EMT-TFs

The main mechanism by which autophagy regulates EMT is the control of EMT-TF degradation. The expression of EMT-TFs is tightly regulated. These TFs are constitutively expressed in the early phases of life (i.e., during embryogenesis) whereas in adult life their expression is strictly regulated and is induced by specific physio-pathologic stimuli. EMT-TFs are the key drivers of tumor metastasis, and their high expression is associated with a poor prognosis in cancer [85]. These factors have generally a brief half-life, and they may undergo degradation by autophagy [133,211].

Twist1 induces the loss of E-cadherin-mediated cell–cell adhesion, facilitating EMT [212]. Zada et al. reported that starvation-induced autophagy led to Snail protein degradation, thus counteracting EMT and limiting the metastatic potential in cancer cells. They also described that Snail was physically associated and colocalized with LC3 and SQSTM1, and that ATG7 knockdown inhibited autophagy-induced Snail degradation [136]. In a similar work, autophagy has been described to degrade Snail under hypoxia conditions in human cardiac microvascular endothelial cells, suggesting a cytoprotective role for autophagy to prevent cardiac fibrosis [213]. According to Grassi et al., in a liver non-tumor cellular system, autophagy may promote the turnover of Snail in a p62/SQSTM1-dependent manner, impairing EMT progression. On the other hand, the authors reported that TGF-β-induced EMT also affects the autophagic flux, suggesting that both processes participate in a complex interplay to regulate the plasticity of hepatocytes [214]. In another study, inhibition of autophagy activated the NF-κB/Snail pathway and induced EMT under the control of SQSTM1/p62 in RAS mutated cancer cells [215]. It was shown that induction of autophagy in neuroblastoma cells may reverse EMT, inhibiting migration and invasion by downregulating Snail and Slug. Conversely, the genetic silencing of BECLIN1, an autophagy initiator, led to upregulation of Snail and Slug and increased migration ability [149].

It has been reported that death-effector domain-containing DNA-binding protein (DEDD), a key effector molecule for cell death signaling receptors, can interact with PI3KC3/Beclin1 to activate autophagy in breast cancer cells, which induces the degradation of Snail and Twist [187]. NOTCH1 intracellular C-terminal domain (NICD) is a transcriptional regulator which takes part in the highly complex transcriptional network regulating EMT mediators, including Snail, ZEB1, and N-cadherin [216] and its inhibition suppresses cancer progression in different models [217,218]. In a very recent study, it has been shown that activating the autophagy lysosomal pathway leads to the degradation of NICD and Snail via physical and functional interactions with SQSTM1/p62 and LC3 in cancer cells. Hence, important EMT mediators such as NICD and Snail may be coordinately regulated by autophagy, and using drugs inducing autophagy could prevent EMT induction as a therapeutic strategy [219].

Thus, in these experimental systems, EMT is negatively regulated by autophagy. As a result, autophagy inhibition can potentially enhance invasiveness and chemoresistance, which is relevant when considering that autophagy induction is a promising tool in cancer therapy.

However, in other experimental conditions, autophagy may promote EMT by affecting the EMT-TFs. BECN1-induced autophagy was shown to accelerate EMT through increased expression of Twist and vimentin, and was suggested with other autophagy modulators as an independent prognostic biomarker in patients with gastric cancer, colorectal carcinoma, and liver cancer [133,220,221]. Several studies have reported that autophagy plays a role in regulating EM T by modulating the activity of EMT-inducing TFs, such as Snail, Twist, and ZEB1. In particular, increased autophagy may favor EMT-TFs activity. Increased acetyl-coenzyme A (acetyl-CoA) generated during autophagy may stabilize Snail by acetylation, facilitating invasion and metastasis of KRAS-LKB1 co-mutated lung cancer cells [222]. Autophagy has also been shown to promote the nuclear localization of Twist, by degrading its negative regulator glycogen synthase kinase 3β (GSK3β) [223,224]. The nuclear localization of Twist promotes the expression of EMT-related genes, leading to the induction of EMT.

Thus, EMT regulation by autophagy is contextual: according to different experimental systems, autophagy may both inhibit or promote the induction of the EMT program (Figure 5).

On the other hand, the EMT process has also been shown to regulate autophagy. The loss of epithelial polarity and the acquisition of a mesenchymal phenotype during EMT can lead to metabolic changes that activate autophagy to maintain cellular homeostasis [151]. It has been reported that EMT-TFs may play a role in regulating the autophagy process. Twist TF, which is a key regulator of EMT, has been shown to induce autophagy by activating the AMPK pathway and inhibiting the mTOR pathway [152]. Moreover, the EMT process can also regulate autophagy by modulating the expression of autophagy master genes. It was demonstrated that under energy stress, Slug is transcriptionally activated by FOXO3 and by interacting with FOXO3 increases the binding affinity of FOXO3 to its response elements and promotes the expression of autophagy-associated genes such as PIK3CA and unc-51-like autophagy activating kinase 1 (ULK1) in Hela cells [225]. However, Snail has been reported to repress the expression of the autophagy-related gene ATG5, leading to the inhibition of autophagy [153].

Overall, autophagy and EMT may interact at multiple crossroads and their interrelation appears to be contextual depending on the specific experimental system analyses. This is a concern when the translational applications of these discoveries are considered (Figure 6).

### 4.4. Clinical Relevance of Interplay between EMT and Autophagy

The exploration of the EMT-autophagy axis in cancer therapy is significantly illuminated by diverse studies, each highlighting the variability in treatment effectiveness tied to this complex interplay. For instance, in colorectal cancer, the intricate relationship between EMT and the emergence of drug-resistant cancer stem-like cells is established, implicating a key molecular axis that governs chemotherapy resistance and metastasis [226]. Additionally, in colon cancer, the role of phosphorylated c-Fos in conferring resistance to 5-Fluorouracil through stem cell-related pathways presents a novel therapeutic target [227]. The potential of metabolism-specific inhibitors to address EMT-driven chemoresistance underscores a strategic complement to conventional treatments [132]. Furthermore, in breast cancer, the link between (N-myc and STAT interactor) NMI expression, autophagy, and enhanced sensitivity to cisplatin through specific molecular pathways exemplifies the nuanced influence of these biological processes on therapeutic outcomes [228]. Recent studies have made significant strides in identifying prognostic biomarkers associated with EMT and autophagy in various cancers. A study on endometrial cancer (EC) developed a four-gene signature (SIRT2, SIX1, CDKN2A, and PGR) for predicting patient prognosis, emphasizing the role of EMT in cancer progression [229]. Similarly, colorectal cancer (CRC) research identified 11 key autophagy-related genes as part of a prognostic model, showcasing the potential of autophagy markers in predicting treatment outcomes [230]. Studies on GBM highlighted Epithelial membrane protein 3 (EMP3) as a significant factor in promoting malignancy and as an independent prognostic factor for patient survival, linking it to the EMT process [231]. In the case of human epidermal growth factor receptor 2 positive (HER2+) breast cancer, Gasdermin B (GSDMB) overexpression was found to be associated with increased resistance to therapy and aggressive tumor behavior, indicating its role in protective autophagy [232]. Emerging research in targeted therapies has demonstrated promising strategies that modulate the EMT-autophagy pathway in cancer treatment. For instance, a study has developed a targeted nano-system that induces ER stress and autophagy in breast cancer, implying that enhancing autophagy can significantly reduce metastasis and improve immune response [233]. Another research on melanoma revealed that alteronol induces apoptosis and inhibits EMT, and its effects are potentiated when combined with an autophagy inhibitor, suggesting a novel treatment approach [157]. Additionally, the complex roles of autophagy in melanoma, especially in the tumor microenvironment, have been highlighted, suggesting careful consideration of autophagy manipulation in cancer therapy [234]. Lastly, a study on gastric cancer found that CD13 inhibition using Ubenimex overcomes cisplatin resistance by suppressing autophagy and EMT, offering a potential new therapeutic strategy [235].

In conclusion, this research underscores a pivotal shift towards personalized cancer treatments, informed by the nuanced interplay between EMT and autophagy. This evolving understanding is crucial for refining cancer therapy, offering prospects for more effective, patient-specific strategies. The significance of these context-dependent associations in various cancers is becoming increasingly clear, highlighting the importance of tailoring treatments to individual EMT-autophagy dynamics. Such an approach could lead to enhanced therapeutic efficacy and reduced side effects, marking a promising direction for future clinical research and strategy development in oncology.

## 5. Conclusions

The intricate interplay between autophagy and EMT has been spotlighted as a paramount regulator in cancer progression, metastasis, and therapeutic resistance. While autophagy traditionally serves as a cell survival mechanism during stress, its dual role in promoting or inhibiting EMT reflects the adaptive and context-dependent nature of tumor cells. This duality is also mirrored in the function of EMT-TFs, which depending on the cellular context, can either promote or inhibit autophagy.

Recent findings have underscored the importance of understanding the reciprocal regulation between EMT and autophagy. The ability of the autophagy pathway to selectively degrade key EMT-related markers, such as E-cadherin, can influence cellular adherence and migration properties. Conversely, EMT-TFs like Twist1, Snail, and Slug can dictate autophagy rates, emphasizing a feedback mechanism that harmonizes cellular transformation and survival processes.

The notion that autophagy can both suppress and facilitate EMT in different settings underscores the importance of context. Factors like cellular environment, tumor type, metabolic status, and specific stimulatory cues can sway the balance in favor of either EMT promotion or inhibition. Similarly, EMT-TFs can act as arbiters, modulating autophagy in response to cellular needs.

Importantly, therapeutic interventions targeting autophagy or EMT require a nuanced understanding of this relationship. While inhibiting autophagy might seem beneficial in contexts where it supports EMT and metastasis, such strategies could inadvertently bolster tumor growth in scenarios where autophagy serves as a brake on EMT. This complexity indicates that patient-specific factors, including tumor type, stage, and genetic makeup, should guide therapeutic decisions.

In summary, the bidirectional relationship between autophagy and EMT represents a dynamic axis in cancer biology. The intricate balance maintained by these processes determines the course of tumorigenesis and metastasis. A deeper, more comprehensive understanding of this relationship promises not only insights into fundamental tumor biology but also offers a roadmap for devising targeted and effective therapeutic strategies. Future research endeavors must prioritize unraveling this relationship in diverse cellular contexts and exploring its therapeutic implications for better patient outcomes.

## 6. Future Directions

Understanding the role of autophagy in the progression of cancer cells remains a pivotal area of exploration. The identification of how autophagy links with EMT in highly proliferative cancer cells could significantly deepen our knowledge about autophagy’s contribution to both the initiation and progression of cancer. This is particularly crucial as specific EMT markers may serve as novel therapeutic targets in cancers where autophagy regulation plays a key role. Future research should focus on identifying these markers and investigating their potential as targets for intervention. This approach could pave the way for more effective, targeted cancer therapies, enhancing our ability to combat this complex and multifaceted disease.

## Figures and Tables

**Figure 1 cancers-16-00807-f001:**
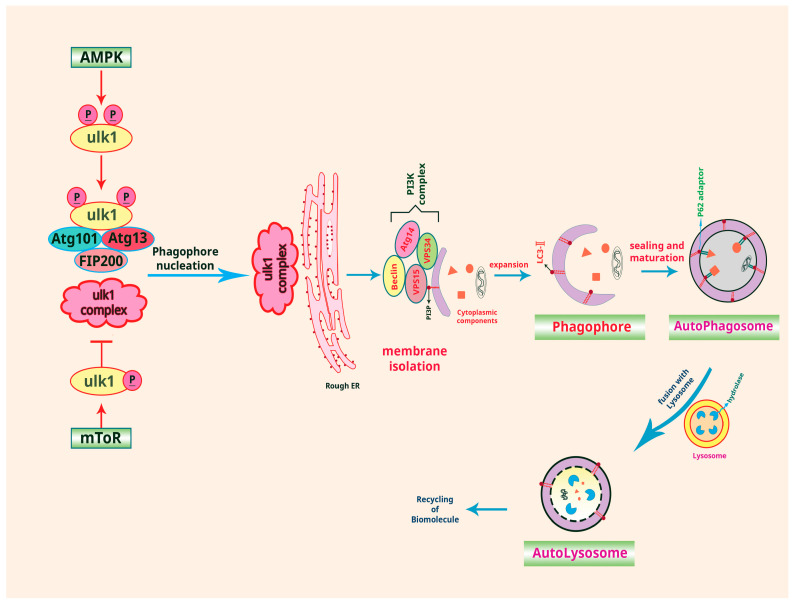
Schematic overview of the macroautophagy process and its key players in induction and nucleation. Macroautophagy involves several distinct stages: Induction: triggered by metabolic stress or treatment, involving the participation of the ULK1 protein complex. Nucleation: characterized by the formation of the phagophore or isolation membrane, primarily stimulated by the PI3K complex. Elongation: where the phagophore expands and develops into the autophagosome. Integration: occurs when the autophagosome containing cytosolic cargo fuses with the lysosome, resulting in the formation of the autolysosome. Degradation: takes place within the autolysosome, as lysosomal hydrolases break down the contents and release primary components back into the cytosol.

**Figure 2 cancers-16-00807-f002:**
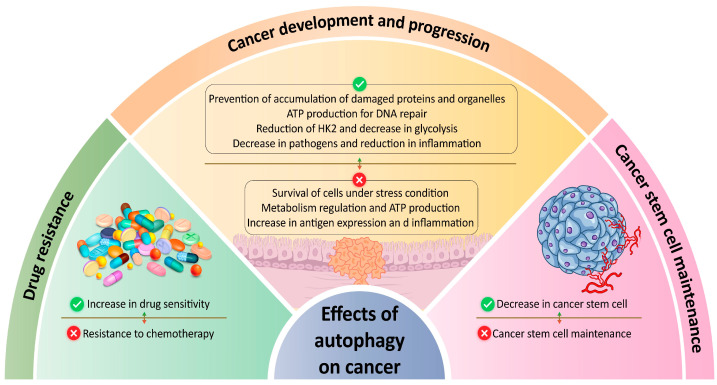
The potential impact of the autophagy mechanism on cancer. The effect of autophagy in cancer is highly contextual and stage-cell specific. Autophagy may eliminate damaged organelles such as mitochondria, thus removing a potential source of new DNA mutations, or alternatively may favor tumor cell survival under stress conditions. Autophagy can exert influence on the development and progression of cancer, as it can either prevent or stimulate cancer development and progression. Additionally, it can affect cancer stem cell maintenance and drug resistance.

**Figure 3 cancers-16-00807-f003:**
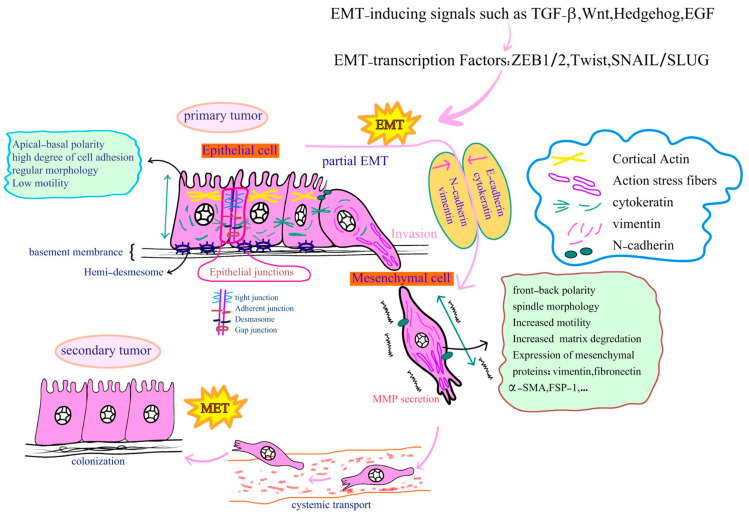
Overview of the EMT/MET process in the metastasis of epithelial cancer cells. Accumulating genetic mutations leads to abnormal cell proliferation and primary tumor formation. Upon activation of specific signaling pathways and transcription factors, epithelial cancer cells undergo epithelial-to-mesenchymal transition (EMT). During this process, epithelial cells lose normal cellular junctions and apical-basal polarity and acquire migratory and invasive properties. Epithelial cells undergoing EMT can enter the lymphatic system or blood vessels, which disseminate them to distant sites. There, they can exit from the circulation and undergo the mesenchymal-to-epithelial transition (MET) process to form secondary tumors (colonization). TGF-β, transforming growth factor-β; EGF, epidermal growth factor; α-SMA, α-smooth muscle actin; FSP-1, fibroblast-specific protein-1; MMP, matrix metalloproteinase.

**Figure 4 cancers-16-00807-f004:**
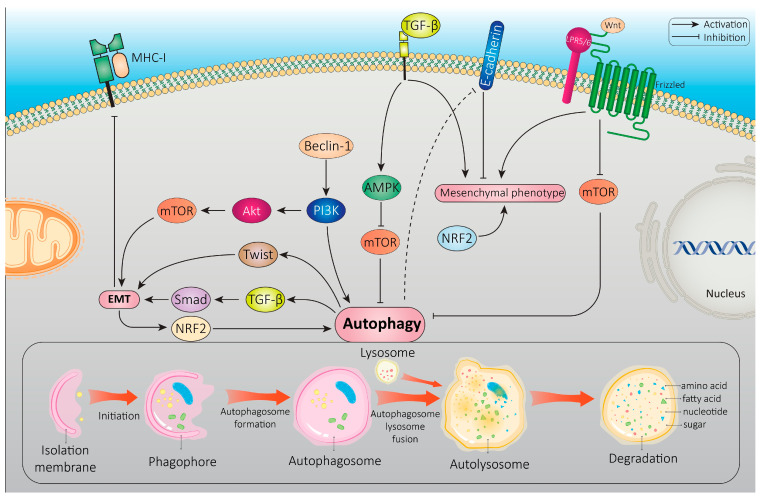
Schematic illustration of the interplay between signaling pathways regulating EMT and autophagy.

**Figure 5 cancers-16-00807-f005:**
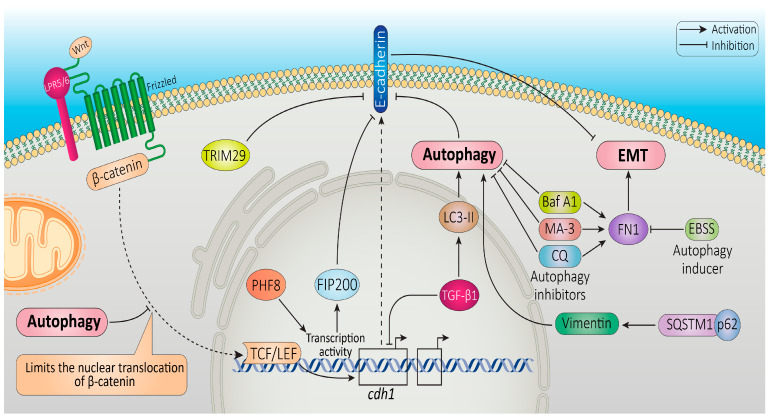
Schematics showing how autophagy might influence EMT and mediators of EMT-related signaling pathways.

**Figure 6 cancers-16-00807-f006:**
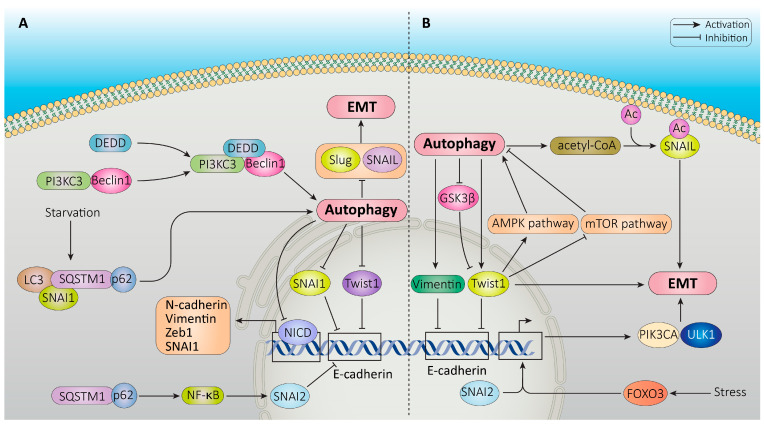
The complex interplay between autophagy and EMT makes their connection context-dependent; (**A**) autophagy induction prevents EMT induction, which is considered as a therapeutic strategy by decreasing the invasiveness and metastatic activity of cancer cells; while on the other hand, (**B**) autophagy may promote EMT by affecting the EMT-inducing transcription factors (EMT-TFs) such as Twist or Snail, and thus could be considered as a cancer biomarker.

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
