# Peer review of "Contribution of Autophagy to Epithelial Mesenchymal Transition Induction during Cancer Progression"

_cancers, 2024, doi:10.3390/cancers16040807_

Round 1

Reviewer 1 Report

Comments and Suggestions for Authors

The present manuscript has high quality and explores important concepts in the field of autophagy. The quality of the Figure 5 could be improved however, to match the others. In overall, the work provides solid references to the scientific community. 

Author Response

Reply: We thank the reviewer 1 for appreciating our work. We choose do not increase the complexity of figure 5 (role of autophagy on EMT) since it partially overlap already with figure 4 (interrelations between autophagy and EMT).

Reviewer 2 Report

Comments and Suggestions for Authors

The manuscript extensively discussed about the importance of autophagy and EMT in cancer development and therapy. It appears far beyond the scope of current title "Autophagy and the fate of cancer cells during epithelial mes-enchymal transition". Much is addressed on the relationship between autophagy and EMT rather than the fate of cancer cells. Therefore, it is advised to modify the title and abstract to make them more relevant to the content.

It would be also benificial to the readers to include the clinical relevance of context-dependent associations between EMT and autophagy, as mentioned in current discussion.

Author Response

Reply: We thank the Reviewer2 for her/his comments. The title and abstract of the paper has been modified in accordance to Reviewer’s 2 concerns. The clinical relevance has now been discussed in section 5.3 “Clinical relevance of interplay between EMT and autophagy”

Reviewer 3 Report

Comments and Suggestions for Authors

The premise of the review is the link between CSCs, EMT and autophagy, which should be interesting as a trinity of causes.

The abstract outlines the role of EMT and also the double-edged sword effect in cancers. Then the link between EMT and autophagy is introduced as changes to morphology, DNA repair system and cell proliferation and differentiation. This is important since these useful links might be useful for assessing therapeutic targets.

The abstract is generally good.

Intro: The intro starts with aspects about cell fate (one of 5 fates) and that regulatory mechanisms govern this decision. The authors could also acknowledge environmental causes in the same sentences (lines 52-3) (I am sure the authors know about this but just to make it clearer for the reader). I also see this is mentioned in the next sentences.

Interestingly, the authors mention the sequence of events for transformation. But is it always the case that there is one causative alteration and then the subsequent alterations are consequential? (lines 58-60) The authors can mention the accumulation of mutations that will confer heightened fitness advantages to the cell and the cell pursues selfish positive selection.

In line 62 it is interesting that the authors mention proliferation and differentiation as two aspects of transformation. The former is always the case but the latter (differentiation) will depend on the cancer type. I assume they refer to cancer cell differentiation, for example, there are varying degrees of differentiation compared to the cell/ tissue of origin (differentiated, differentiating, poor-diffenritated in neuroblastic cancers). In addition, in some cancers such as ALCL the cell of origin is a bit of a conundrum. My point is whether the authors are confident that they should use differentiation as one of the features of cancer cells.

Then the authors mention some of the cancer hallmarks. It would be useful to refer to them as those.

Then the authors mention endogenous and exogenous carcinogens.

Then the topic of DNA damage is mentioned, the authors could talk about ATM/ ATR/ chk2/chk1. Then this is linked to apoptosis and autophagy.

The authors could formulate their aims at the end of section 1.

Chapter 2:

The general process of macroautophagy (2.1) can be shown in a figure.

The authors could refer to the activatory and inhibitory phosphorylation marks for ULK1 bestowed by AMPK and mTOR, respectively (lines 98-100). Please give 1 example for TFEB and FOXO1 targets.

In the next paragraph, the authors could mention the early and late cancer-stage effects of autophagy.

Evidence suggests that autophagy is closely linked to the DDR network, 123

and that DNA damage can stimulate autophagy in both normal and abnormal situations. The authors could elaborate on what they mean by normal and abnormal situations.

The topics in lines 118-126, there are quite a few sentences that are abstract but I feel they could be more detailed. Please elaborate on these two sentences:

For the DNA repair machinery to function properly, cells require an efficient 125

autophagic process. Furthermore, autophagy helps maintain bioenergetic fitness by 126

providing metabolic precursors that generate ATP required for optimal DNA repair and 127

maintaining nucleotide homeostasis for DNA synthesis.

The authors mention the link between excessive DNA damage that can lead to autophagy-induced cell death or cell cycle arrest. But the example they give is about an autophagy gene (BECLIN1) deletion leading to genomic instability. 

The next paragraph talks about inflammation in cancer which is also linked to autophagy. I am not quite sure what the logical order of topics is, to be honest. The authors could add additional subsections to consolidate similar topics.

The link to metabolism is interesting. What is the impact of modulating Hexokinase 2 in normal cells (as an off-target effect)?

CSCs and cancer-initiating cells are not the same population. The authors could mention this in line 177. Also, treatment resistance cells are usually referred to as persister cells

Some interesting links between CSCs and the role of autophagy in maintaining them are mentioned.

What is the mechanism of the study mentioned in lines 183-6? By what mechanism does autophagy increase resistance to chemotherapy?

How does autophagy maintain stemness in the Shi study? Is it regulating NANOG, OCT4, SOX2?

For the readers to benefit more from example 35, the authors could talk about clonogenicity as one of the properties of CSCs earlier in this section. Also, other important aspects of CSCs are tumour propagation in vivo (in an already established cancer by cancer-initiating cells) and also efflux of toxic compounds and chemotherapeutics. The authors could also mention some of the markers of CSCs.

The sections could benefit from tables that summarise key points about each aspect. For example, autophagy is a therapeutic target or link to CSCs. Chapter 2 can be summarised in one table. 

The link between ATG9B expression and drug resistance can be elaborated, how does this autophagy gene exactly increase resistance? You have mentioned what regulates it and that it is expressed in resistance gliomas but how exactly does ATG9B execute its role?

The same question about TLR9. The mechanism of BRCA1-mediated drug resistance is clearer since it is directly linked to DNA damage repair and could maintain stemness.

The clinical trials are interesting.

The same comment about tables applies to sections 3 and 4.

Section 3. The EMT definition in lines 302-304 is eloquent.

The section on morphological changes is quite comprehensive. It starts with physiological tissue organisation and various adhesion molecules and junctions. Then mesenchymal cells and their properties are mentioned in great detail. The EMT is defined and elaborated on. Finally, important signalling pathways and players are introduced. 

The authors could however distinguish between EMT for physiological and pathological processes. For example, wound healing uses EMT but it is a physiological process. Migration also takes place during embryological development such as the migration of neural crest cells from the neural tube across a few main paths. So, it is important to note the differences between signalling pathways and players that are involved in physiological versus pathological processes (including EMT in cancer).

Amoeboid migration is characterized by membrane blebbing, weak adhesion or pushing movements, high myosin II activation, and rapid motility. On the other hand, mesenchymal migration involves ECM degradation, strong adhesions, front-rear polarization, and the formation of actin-rich membrane protrusions like lamellipodia and filopodia. This is interesting but does the former occur in cancers as well? The authors mention these are interchanged which begs more explanation. 

The next paragraph talks about EMT and invasion as an inseparable aspect of metastasis and then EMT and protein hydrolysis and this is supported by a figure.

Figure Legend 2 is only a title and needs to be substantiated. 

The content of 3.3 could be summarised in a table, as mentioned before.

The topic on CSC and MET in line 511 is interesting but begs the question of why the expression of Wnt, HH and Notch are linking CSCs to EMT. This is not clear. What markers are you referring to inline 512? The authors divert to CSCs expressing ABC transporters but how is this relevant to the sentences above it that talk about EMT/ CSCs and 3 signalling pathways? This needs more detail.

The next 2 topics (TME and resistance) are okay but again the flow of topics is not clear. Why have the authors decided on these sequences of topics here? It may become clearer with an additional subsection.

Also, this section could have a figure.

Section 4. A higher figure frequency is given in this section.

I particularly liked section 4.1. since it is informative. This content could be placed/ summarised in a table as well. The same applies to 4.2-3.

Figure legends for figures 3-5 also need substantiation. Overall, these 3 figures have been the highlight of the study. I appreciate the time and effort this demands.

Comments on the Quality of English Language

Minor editing

Author Response

The premise of the review is the link between CSCs, EMT and autophagy, which should be interesting as a trinity of causes. The abstract outlines the role of EMT and also the double-edged sword effect in cancers. Then the link between EMT and autophagy is introduced as changes to morphology, DNA repair system and cell proliferation and differentiation. This is important since these useful links might be useful for assessing therapeutic targets.The abstract is generally good.

Reply: We thank the Reviewer 3 for her/his general appreciation of our work.

Intro: The intro starts with aspects about cell fate (one of 5 fates) and that regulatory mechanisms govern this decision. The authors could also acknowledge environmental causes in the same sentences (lines 52-3) (I am sure the authors know about this but just to make it clearer for the reader). I also see this is mentioned in the next sentences

Reply: The first part of the introduction has been re-written and environmental causes have now been clearly introduced.

Interestingly, the authors mention the sequence of events for transformation. But is it always the case that there is one causative alteration and then the subsequent alterations are consequential? (lines 58-60) The authors can mention the accumulation of mutations that will confer heightened fitness advantages to the cell and the cell pursues selfish positive selection.

Reply: The concept of accumulation of mutations has been clearly stated in line 60: “This transformation entails the progressive accumulation of DNA mutations in cancer-related genes”.

In line 62 it is interesting that the authors mention proliferation and differentiation as two aspects of transformation. The former is always the case but the latter (differentiation) will depend on the cancer type. I assume they refer to cancer cell differentiation, for example, there are varying degrees of differentiation compared to the cell/ tissue of origin (differentiated, differentiating, poor-diffenritated in neuroblastic cancers). In addition, in some cancers such as ALCL the cell of origin is a bit of a conundrum. My point is whether the authors are confident that they should use differentiation as one of the features of cancer cells.

Then the authors mention some of the cancer hallmarks. It would be useful to refer to them as those

Reply: We general consider the dedifferentiation process as typical of cancer transformation, although cancer cells may exhibit high grade of differentiation. This is stated in this sentence:  “Cancer cells emerge as transformed derivatives of normal cells, exhibiting an extraordinary capacity for rapid proliferation and dedifferentiation.”

Then the authors mention endogenous and exogenous carcinogens. Then  the topic of DNA damage is mentioned, the authors could talk about ATM/ ATR/ chk2/chk1. Then this is linked to apoptosis and autophagy.

The authors could formulate their aims at the end of section 1.

Reply: ATM-ATR and Checkpoint kinases are now discussed in lines 87-89.

Chapter 2:

The general process of macroautophagy (2.1) can be shown in a figure.

Reply: As suggested by Reviewer 3, a new figure describing macroautophagy (Fig. 1) has been included.

.

The authors could refer to the activatory and inhibitory phosphorylation marks for ULK1 bestowed by AMPK and mTOR, respectively (lines 98-100). Please give 1 example for TFEB and FOXO1 targets.

In the next paragraph, the authors could mention the early and late cancer-stage effects of autophagy.

Evidence suggests that autophagy is closely linked to the DDR network, 123

and that DNA damage can stimulate autophagy in both normal and abnormal situations. The authors could elaborate on what they mean by normal and abnormal situations.

Reply: A section has been added that clarifies the influence of mTOR and AMPK on ULK. Additionally, targets for TFEB and FOXO1 have been included. Further information on the early and late cancer-stage effects of autophagy has been discussed in this section. The statement on normal and abnormal situations have been corrected by replacing by pathological and non-pathological conditions.

The topics in lines 118-126, there are quite a few sentences that are abstract but I feel they could be more detailed. Please elaborate on these two sentences:

For the DNA repair machinery to function properly, cells require an efficient autophagic process. Furthermore, autophagy helps maintain bioenergetic fitness by

providing metabolic precursors that generate ATP required for optimal DNA repair and

maintaining nucleotide homeostasis for DNA synthesis.

Reply: As suggested by the reviewer, additional information on the DNA machinery have been incorporated into this section.

The authors mention the link between excessive DNA damage that can lead to autophagy-induced cell death or cell cycle arrest. But the example they give is about an autophagy gene (BECLIN1) deletion leading to genomic instability.

Reply: The example given about an autophagy gene (BECLIN1) deletion which leads to genomic instability has been re-phrased to match with the preceding sentence, where autophagy's role in DNA repair through ATP provision was discussed. Specifically, we cited an instance illustrating that a deficiency in an autophagy-related gene, BECLIN1, can result in genomic instability in an ovarian cancer model. As suggested by the reviewer we have amended the paragraph on inflammation in cancer.

The link to metabolism is interesting. What is the impact of modulating Hexokinase 2 in normal cells (as an off-target effect)?

Reply: In response to the comment on the impact of modulation Hexokinase 2 in normal cells, that in the cited study, the effects of modulating Hexokinase 2 were evaluated only in tumor cells.

CSCs and cancer-initiating cells are not the same population. The authors could mention this in line 177. Also, treatment resistance cells are usually referred to as persister cells.

Reply: The comment on Cancer Stem Cells (CSCs) and cancer-initiating cells has been addressed in the revised paper.

Some interesting links between CSCs and the role of autophagy in maintaining them are mentioned

What is the mechanism of the study mentioned in lines 183-6? By what mechanism does autophagy increase resistance to chemotherapy?

Reply: The answer to the question about the role of autophagy in resistance to chemotherapy has been addressed in the revised manuscript. Perhaps, autophagy aids CSCs in resisting chemotherapy and radiation therapy by facilitating DNA repair and mitigating oxidative stress.

How does autophagy maintain stemness in the Shi study? Is it regulating NANOG, OCT4, SOX2?

Reply: The answer to the question about the regulatory role of autophagy on NANOG, OCT4, SOX2 is that knocking down TINCR can result in reduced expression of transcription factors POU5F1, SOX2, Nanog, and surface marker CD44.

For the readers to benefit more from example 35, the authors could talk about clonogenicity as one of the properties of CSCs earlier in this section. Also, other important aspects of CSCs are tumour propagation in vivo (in an already established cancer by cancer-initiating cells) and also efflux of toxic compounds and chemotherapeutics. The authors could also mention some of the markers of CSCs.

Reply: Further information about other aspects of CSCs has been incorporated in this section.

The sections could benefit from tables that summarise key points about each aspect. For example, autophagy is a therapeutic target or link to CSCs. Chapter 2 can be summarised in one table.

Reply: As suggested by reviewer the content of Chapter-2 has been condensed and presented in Figure 2.

The link between ATG9B expression and drug resistance can be elaborated, how does this autophagy gene exactly increase resistance? You have mentioned what regulates it and that it is expressed in resistance gliomas but how exactly does ATG9B execute its role?

Reply: Information regarding the role of ATG9B has been incorporated. In this connection, it has been reported that miR-181a expression leads to an elevation in autophagy by increasing ATG5 and ATG2B levels. According to this report As ATG5 and ATG2B play crucial roles in the early formation of autophagosomes and their increase results in an overall enhancement of autophagy. Consequently, the heightened autophagy levels leads to a reduction in stem cell markers, including OCT4 and SOX2.

The same question about TLR9.

Reply: Our answer to the question about TLR9 is that our goal was to explain the correlation between autophagy and drug resistance. The study was to better understanding the mechanisms responsible for resistance to Sorafenib through Toll-like Receptor 9 (TLR-9), uncovering a correlation among drug resistance, TLR-9, and various markers and mechanisms. Information given about the correlation between TLR-9, autophagy and drug resistance gives further insight into the impact of TLR-9 on autophagy and drug resistance.

Please note that the similarity of the text has been checked and significantly reduced in the revised manuscript. Also all other comments raised by reviewers have been addressed in the manuscript.

Section 3. The EMT definition in lines 302-304 is eloquent.

The section on morphological changes is quite comprehensive. It starts with physiological tissue organisation and various adhesion molecules and junctions. Then mesenchymal cells and their properties are mentioned in great detail. The EMT is defined and elaborated on. Finally, important signalling pathways and players are introduced.

We thank the Reviewer3 for appreciating our description of EMT.

The authors could however distinguish between EMT for physiological and pathological processes. For example, wound healing uses EMT but it is a physiological process. Migration also takes place during embryological development such as the migration of neural crest cells from the neural tube across a few main paths. So, it is important to note the differences between signalling pathways and players that are involved in physiological versus pathological processes (including EMT in cancer).

Reply: We have clearly stated in the introduction that EMT has both a physiological and pathological conditions. Due to the nature of the present review, we only focused on the pathological part.

Amoeboid migration is characterized by membrane blebbing, weak adhesion or pushing movements, high myosin II activation, and rapid motility. On the other hand, mesenchymal migration involves ECM degradation, strong adhesions, front-rear polarization, and the formation of actin-rich membrane protrusions like lamellipodia and filopodia. This is interesting but does the former occur in cancers as well? The authors mention these are interchanged which begs more explanation.

Yes, both ameboid and collective migration have been demonstrated in cancer cells.

The next paragraph talks about EMT and invasion as an inseparable aspect of metastasis and then EMT and protein hydrolysis and this is supported by a figure. Figure Legend 2 is only a title and needs to be substantiated.

Reply: This has been dealt with in e understand that increasing the number of figures/tables could be beneficial for a further understanding of our topic, but we believe that we have reached a sufficient level of detail in this version of the manuscript.

The content of 3.3 could be summarised in a table, as mentioned before.

The topic on CSC and MET in line 511 is interesting but begs the question of why the expression of Wnt,HH and Notch are linking CSCs to EMT. This is not clear. What markers are you referring to inline 512? The authors divert to CSCs expressing ABC transporters but how is this relevant to the sentences above it that talk about EMT/ CSCs and 3 signalling pathways? This needs more detail.

We now added some details of molecular markers induced by  WNT and HH pathway in CSC. Moreover, we clarified that ABC transporter expression in CSCs is under control of NF-kB and SREBP2 transcription factors, as well as by EMT TFs TWIST and SNAIL.

The next 2 topics (TME and resistance) are okay but again the flow of topics is not clear. Why have the authors decided on these sequences of topics here? It may become clearer with an additional subsection.

Also, this section could have a figure.Section 4. A higher figure frequency is given in this section.

I particularly liked section 4.1. since it is informative. This content could be placed/ summarised in a table as well. The same applies to 4.2-3.

Figure legends for figures 3-5 also need substantiation. Overall, these 3 figures have been the highlight of the study. I appreciate the time and effort this demands.

Reply: We understand that increasing the number of figures/tables could be beneficial for a further understanding of our topic, but we believe that we have reached a sufficient level of detail in this version of the manuscript.

Reply:  In this revised version, New sections, i.e. section 5. Clinical relevance and section 7. Future directions, have been added to the manuscript. New references have been inserted in the paper according to the modification made on the manuscript.

Reviewer 4 Report

Comments and Suggestions for Authors

Manuscript is well paced. However, should clearly highlight the novelty and the need of the review.

Comments on the Quality of English Language

Language is fine. 

Author Response

Reply: We thank the Reviewer 4 for her/his appreciation of our work. Although there is plenty of papers dealing on the role of EMT and autophagy alone in cancer transformation, the multiple relations between these two physio-pathological processes are much less analyzed and are extremely relevant in a translational perspective. We clearly state in the abstract that this is the focus of this review article.

Round 2

Reviewer 3 Report

Comments and Suggestions for Authors

The authors have addressed my comments